# Path Counting on Tree-like Graphs with a Single Entropic Trap: Critical Behavior and Finite Size Effects

**DOI:** 10.3390/e25091318

**Published:** 2023-09-09

**Authors:** Alexey V. Gulyaev, Mikhail V. Tamm

**Affiliations:** 1Independent Researcher, 119234 Moscow, Russia; sept1599@yandex.ru; 2CUDAN Open Lab and School of Digital Technologies, Tallinn University, 10120 Tallinn, Estonia

**Keywords:** path counting, maximal entropy random walk, localization transition, critical slowdown

## Abstract

It is known that maximal entropy random walks and partition functions that count long paths on graphs tend to become localized near nodes with a high degree. Here, we revisit the simplest toy model of such a localization: a regular tree of degree *p* with one special node (“root”) that has a degree different from all the others. We present an in-depth study of the path-counting problem precisely at the localization transition. We study paths that start from the root in both infinite trees and finite, locally tree-like regular random graphs (RRGs). For the infinite tree, we prove that the probability distribution function of the endpoints of the path is a step function. The position of the step moves away from the root at a constant velocity v=(p−2)/p. We find the width and asymptotic shape of the distribution in the vicinity of the shock. For a finite RRG, we show that a critical slowdown takes place, and the trajectory length needed to reach the equilibrium distribution is on the order of $\sqrt{N}$ instead of $\log_{p-1}N$ away from the transition. We calculate the exact values of the equilibrium distribution and relaxation length, as well as the shapes of slowly relaxing modes.

## 1. Introduction

One of the most natural ways to characterize a graph or a network is by its adjacency and Laplace matrices [1,2]. The adjacency matrix of an *N*-node network is defined as N×N matrix A, where the entry $A_{ij}=1$ if nodes numbered *i* and *j* are connected to each other, and 0 otherwise. The elements of the Laplacian matrix L of a graph are defined as
(1)$$L_{ij}=deg_{j}\delta_{ij}-A_{ij}$$
where $deg_{j}$ is the degree of node *j*. This matrix defines the relaxation of a continuous-time diffusion process on a graph, i.e., the probability distribution to find a particle at one of the nodes, $P_{L}\left(t\right)={\left(P_{L}\left(1,t\right),P_{L}\left(2,t\right),\ldots ,P_{L}\left(N,t\right)\right)}^{T}$, evolves according to the following:(2)$${\dot{P}}_{L}=LP_{L}$$
and, therefore, eigenvalues and eigenvectors of the Laplacian play a crucial role in describing the relaxation of a simple diffusion process on the networks (see, e.g., [3]).

It seems natural to ask whether there is a process whose relaxation is related in a similar way to the eigenvalues and eigenvectors of the adjacency matrix. We define a sequence of vectors $Z\left(t\right)={\left(Z_{1}\left(t\right),Z_{2}\left(t\right),\ldots ,Z_{N}\left(t\right)\right)}^{T}$ that are dependent on discrete time *t*, satisfying
(3)Z(t+1)=AZ(t).Appended with the initial condition $Z\left(0\right)={\left(1,1,\ldots ,1\right)}^{T}$, it can be interpreted as a vector of partition functions counting trajectories of length *t* on a graph, with $Z_{i}\left(t\right)$ counting trajectories ending at the *i*-th node. Moreover, we define
(4)$$P_{A}\left(i,t\right)=Z_{i}\left(t\right)/\sum_{j}Z_{j}\left(t\right)$$
which is the probability distribution to find the end of a randomly chosen trajectory at point *i*. A similar problem arises (although typically in a continuous space setting) in the classical polymer theory [4,5,6], so the path-counting problem (PC) described above is also often referred to as the counting of “conformations of an ideal polymer” [7,8]. Note the difference between the probability distribution (Equation 4) and that of an endpoint of a symmetric discrete-time random walk on a graph (Equation 2). Indeed, in a symmetric random walk, the probabilities of trajectories are weighted inversely to the product of the degrees of the visited nodes, while in the path-counting problem (PC), all trajectories are weighted equally.

Is it possible to reformulate a path-counting problem as a random walk problem, i.e., as a process satisfying
(5)$$P_{A}\left(t+1\right)=TP_{A}\left(t\right)$$
with some stochastic transfer matrix T? To the best of our knowledge, the answer is “no”, but for finite graphs, a very close approximation known as the maximal entropy random walk (MERW) is possible [9,10]. Namely, if we define $\lambda_{1},\Phi_{1}$ to be the largest eigenvalue of the adjacency matrix and the corresponding eigenvector, and choose the elements of the transfer matrix as
(6)$$T_{ij}=\lambda_{1}^{-1}A_{ji}\frac{\phi_{1i}}{\phi_{1j}},$$
where $\phi_{1i}$ is the *i*-th element of $\Phi_{1}$, then (i) these weights can be understood as step probabilities of some random walk (indeed,
(7)$$\sum_{i}A_{ji}\phi_{1i}=\lambda_{1}\phi_{1j}$$
so the probability is conserved), and (2) the trajectory weight in such a walk is only dependent on its length, starting point, and endpoint, i.e., all trajectories with given lengths and endpoints are equiprobable. As a result, if one considers sufficiently long MERW trajectories, the density of links at a given site *i* becomes proportional to the square of the value of the largest eigenvector at this site $\phi_{1i}^{2}$, in full analogy with the density of the ideal polymer in the external field [4,6].

Thus, on the one hand, MERW and PC problems are fundamentally related. Answers to both can be fully formulated in terms of eigenvectors and eigenvalues of the same adjacency matrix A. On the other hand, the probability distribution $P_{A}\left(t\right)$ (probability distribution of the endpoint of a trajectory of a given length) defined above for the PC problem has no direct analog in MERW. We formulate our results below in terms of this probability distribution, so, to avoid confusion, we restrict ourselves to the terminology of the path-counting problem, although the results can easily be reformulated in terms of MERW.

The most striking feature of the PC and MERW as opposed to regular random walks is the potential for localization. Indeed, while on any finite connected graph, $P_{L}$ converges to a uniform distribution for sufficiently long trajectories, it is not generally the case for $P_{A}$. In fact, large-degree nodes work as entropic traps in the PC problem, which may lead to the localization of trajectories. Perhaps the simplest system where such localization can happen is a regular tree with a branching degree *p* and a single “special” node (“root”) with a different branching degree $p_{0}$. This system has been studied extensively in [8] (also see recent generalizations in [11,12]) where it has been shown that for an infinite tree, there exists a critical branching degree:(8)$$p_{0}=p_{cr}=p\left(p-1\right),$$
such that for root degrees $p_{0}$ that are smaller than $p_{cr}$, sufficiently long trajectories span the whole graph, and all nodes are visited with approximately equal probabilities; for larger values of $p_{0}$, trajectories are localized in the vicinity of the root, causing the probability of identifying the endpoint of a randomly selected trajectory at a distance *x* from the root to decrease exponentially with *x*.

Interestingly, in the analogous MERW problem, which has been studied in [13], the localization transition occurs at a different value of the root degree:(9)$$p_{0}=p_{cr}^{MERW}=2\left(p-1\right).$$This discrepancy can be understood when recognizing that the large-*t* distributions of trajectory inner points and their ends, typically studied in the MERW and PC setups, are proportional to $\phi_{1i}^{2}$ and $\phi_{1i}$. It is clear from the symmetry of the system that values of $\phi_{1i}$ can only depend on distance *x* from node *i* to the root, $\phi_{1i}=\phi_{1x}$, and the probability of being at distance *x* from the origin is proportional, up to the normalization constant to $\phi_{1x}^{2}{\left(p-1\right)}^{x}$ and $\phi_{1x}{\left(p-1\right)}^{x}$ for inner points and endpoints, respectively. As a result, localization for endpoints occurs if $\phi_{1x}$ decays with *x* faster than ${\left(p-1\right)}^{-x}$. For inner points, localization occurs if the decay is faster than ${\left(p-1\right)}^{-x/2}$.

Notably, the standard Brownian random walk on a tree-like graph can be localized around a single site, as discussed in [14], but for this to happen, one needs to apply a uniform global external field [14]. Indeed, a random walk on a tree-like graph can be mapped on a biased random walk on a half-line [15], and if the external field is strong enough to compensate for that bias [16], condensation occurs in a similar way to a random walk on a half-line that is biased in the direction of the origin. Contrary to that, localization in the path-counting problem occurs as a result of a point-like local disorder [8] or, equivalently, as noted in [17,18], as a result of an external field applied only to the endpoint of the trajectory. For Brownian motion, such localization is impossible; it is easy to see that in a potential composed of a localized well at the origin and a constant bias steering away from it, the Brownian walker always escapes.

In this paper, we revisit this system and pay special attention to the behavior at the transition point $p_{0}=p_{cr}$. First, we consider the behavior on an infinite tree. Here, we prove the results presented in [8], asserting that the distribution of the endpoint *x* of the trajectory commencing at the root asymptotically has the form of a traveling wave
(10)$$P_{A}\left(x,t\right)∼\frac{1}{vt}\Theta \left(vt-x\right)$$
where velocity *v* equals (p−2)/p and Θ(x) is the Heaviside theta function. We calculate the shape of the wave in the vicinity of the front, i.e., for $vt-x∼O(\sqrt{t})$. Second, we study the critical behavior of trajectories for $p_{0}=p_{cr}$ on a finite locally tree-like graph with a fixed degree of all nodes, except the root (random regular graph, RRG). In particular, we find the limiting distribution $P_{A}\left(x,t\to \infty \right)$ and show that this limiting distribution is approached extremely slowly, so that the maximal relaxation time $t_{relax}∼\sqrt{N}$ as opposed to $t_{relax}∼\log_{\left(p-1\right)}N$ for $p_{0}\neq p_{cr}$.

## 2. Model and Research Questions

Following [8], let us briefly review what is known about path localization on an infinite regular tree with a single special point. We consider an infinite tree graph with coordination number *p* in all points, except for a single point (root) with a different coordination number $p_{0}>p$. We are interested in the statistics of long paths on such a graph. On an infinite tree, the parity of x+t, where *x* is the distance of the end of a *t*-step trajectory to the root, is exactly conserved. For the statistics of the endpoints of long paths, three regimes are possible [8].

If $p_{0}<p_{cr}$, the typical trajectory escapes from the vicinity of the root, with an average distance from the trajectory’s endpoint to the root growing as 〈x(t)〉=(p−2)t/p. The fluctuation of x(t) in this case is normally distributed with the width of distribution proportional to $\sqrt{t}$. This behavior is completely analogous to the behavior of a trajectory on the infinite regular tree (corresponding to $p_{0}=p$), and can be described as a biased discrete-time Brownian motion on a half-line.

In turn, if $p_{0}>p_{cr}$, localization occurs, meaning that the endpoint of a long trajectory is localized in the vicinity of the root with 〈x(t)〉 converging to a constant for the large *t*, and the probability of finding the trajectory’s endpoint at distance *x* away from the root diminishing exponentially with *x*.

Finally, in a critical case, $p_{0}=p_{cr}$, very interesting behavior is numerically observed in [8]: the probability of finding the end of an *t*-step trajectory starting from the root at a distance *x* from it is roughly a step function:(11)$$P\left(x,t\right)\approx \left\{\begin{array}{cc} \frac{2p}{p-2}t^{-1} & \;\mathrm{for}\;x\ll \frac{(p-2)t}{p}\;\mathrm{and}\;\mathrm{even}\;x+t,\\ 0 & \;\mathrm{otherwise},\; \end{array}\right.$$
with some crossover function linking these limiting regimes for x≈(p−2)t/p (see Figure 1b). However, this latter result has not been proven in [8].

In this paper, we focus on the details of the critical behavior, i.e., the path statistics in a tree with branching degrees *p* in all points but the root, where the branching degree is $p_{0}=p_{cr}=p\left(p-1\right)$ in order to further elucidate the behavior in this regime. First, in Section 3, we address the behavior of the infinite system and prove (Equation 11). In Section 4, we discuss how the critical regime looks in the case when a tree is large but finite. We show that it is important to distinguish between a Cayley tree (a regular tree with a finite fraction of boundary nodes of degree one) in the limit of a large number of generations and a random regular graph, i.e., a local tree-like graph where nodes far away from the root are randomly connected to guarantee that each node—except for the root—has degree *p*. In the first system, there is no critical behavior at $p=p_{cr}$ for sufficiently long trajectories, while for the second system, a very peculiar ultra-slow relaxation occurs.

## 3. Critical State on an Infinite Tree

Introduce a partition function $Z_{t}\left(x\right)$, counting the number of paths beginning at the root, consisting of *t* steps and ending at a distance *x* from the root. It is easy to see that it satisfies:(12)$$\left\{\begin{array}{cc} Z_{t+1}\left(x\right)=\left(p-1\right)Z_{t}\left(x-1\right)+Z_{t}\left(x+1\right) & \;\mathrm{for}\;x>1\\ Z_{t+1}\left(x\right)=p_{0}Z_{t}\left(x-1\right)+Z_{t}\left(x+1\right) & \;\mathrm{for}\;x=1\\ Z_{t+1}\left(x\right)=Z_{t}\left(x+1\right) & \;\mathrm{for}\;x=0 \end{array}\right.$$
with initial condition $Z_{0}\left(x\right)=\delta_{x,0}$, and we are particularly interested in $p_{0}=p_{cr}$. The introduction of a renormalized partition function $W_{t}\left(x\right)$ according to
(13)$$Z_{t}\left(x\right)={\sqrt{p-1}}^{t+x}W_{t}\left(x\right)$$
allows us to symmetrize the equations, giving
(14)$$\left\{\begin{array}{cc} W_{t+1}\left(x\right)=W_{t}\left(x-1\right)+W_{t}\left(x+1\right)+\left(p-1\right)\delta_{x,1}W_{t}\left(x-1\right) & x\geq 0\\ W_{t}\left(x\right)=0 & x<0\\ W_{t=0}\left(x\right)=\delta_{x,0} \end{array}\right.$$
where we substitute the critical value of *p* according to (Equation 8). These equations define, for non-negative integer indices *t* and *x*, a two-parametric sequence *W*, which has a simple combinatorial meaning. Indeed, for p=1 it simply enumerates trajectories in the upper-right quadrant, consisting of right-up and right-down steps (see Figure 2), which start at the origin and end at the point with coordinates (t,x). For arbitrary *p*, it enumerates *weighted* trajectories:(15)$$W_{t}\left(x\right)=\sum_{\mathrm{traj}.\;\mathrm{from}\;\left(0,0\right)\;\mathrm{to}\;(t,x)}p^{\#\mathrm{visits}\;\mathrm{of}\;x=0}=\sum_{m}C_{t}^{m}\left(x\right)p^{\left(m+1-\delta_{x,0}\right)},$$
where we introduce $C_{t}^{m}\left(x\right)$—the number of trajectories from (0,0) to (t,x)—which return to the horizontal axis x=0 exactly *m* times. Note that the trajectory acquires weight *p* every time it leaves x=0, so the additional 1 in the power on the left-hand side accounts for the initial step, and the Kronecker symbol $\delta_{x,0}$ accounts for the scenario where, for x=0, the trajectory is still on its last return without having left 0.

To calculate $C_{t}^{m}\left(x\right)$, note that there is a bijection between all paths going from (0,0) to (t,x) that visit the horizontal axis *m* times, and all paths from (0,0) to (t−m,x+m) that never visit the horizontal axis; therefore,
(16)$$C_{t}^{m}\left(x\right)=C_{t-m}^{0}\left(x+m\right).$$This bijection is constructed as follows. Consider first a path from (0,0) to (t,x) that intersects the horizontal axis *m* times and exclude from it all the steps from x=1 to x=0 (marked in red in Figure 2); shift the remaining parts of the path upward and to the left so that they form a continuous path. Clearly, the resulting path goes from (0,0) to (t−m,x+m) without returning to the horizontal axis. Consider now an arbitrary path from (0,0) to (t−m,x+m) that does not visit the horizontal axis; for each vertical coordinate value from i=1 to i=m, mark a point $\left(t_{i},i\right)$ on the path where it is attained for the final time. It follows from x+m≥m that these points exist and form a strictly increasing sequence $t_{1}<t_{2}<\ldots <t_{m}$. Now, split the path in these points, shift each of the resulting subpaths by vector (i,−i), and insert downward steps connecting these paths together, as shown in Figure 2. Clearly, the path resulting from this transformation is a path from (0,0) to (t,x), visiting the horizontal axis exactly *m* times. It is easy to see that in both the direct and reverse transformations, different initial paths result in different transformed paths; thus, there exists a one-to-one correspondence between these two path classes, proving (Equation 16). In turn, the number of paths not visiting zero $C_{t-m}^{0}\left(x+m\right)$ can be easily calculated, e.g., by the method of images, with the result being
(17)Ctm(x)=Ct−m0(x+m)==t−m−1[(t−m−1)+(x+m−1)]/2−t−m−1[(t−m−1)+(x+m+1)]/2==t−m−1(t+x)/2−1−t−m−1(t+x)/2=x+mt−mt−m(t+x)/2.Substituting this result into (Equation 13) and (Equation 15), we obtain the following:(18)Zt(x)=(p−1)t+x∑m=0(t−x)/2x+mt−mt−m(t+x)/2pm+1−δ0,x.Consider now the asymptotic behavior of this expression. Assume that t,x≫1 and x/t=α. Then, after replacing factorials with their approximate values using the Stirling formula and introducing notation,
(19)β=(1−α)/2,ξ=m/(tβ),
obtain
(20)t−m(t+x)/2≈12πt1−ξββ(1−β)(1−ξ)β(1−ξ)1−βt(1−β)1−ξββ(1−ξ)t(1−ξβ).Substituting this expression into (Equation 18), rewrite the partition function in the following approximate form:(21)$$Z_{t}\left(\alpha t\right)\approx \beta p\sqrt{t}\;{\left(p-1\right)}^{t(1-\beta )}\int_{0}^{1}\psi \left(\xi \right)e^{t\phi (\xi )}d\xi ,$$
where
(22)$$\phi \left(\xi \right)=\xi \beta \ln p-\left(1-\xi \beta \right)\ln\left(\frac{\beta (1-\xi )}{1-\xi \beta }\right)-\left(1-\beta \right)\ln\left(\frac{1-\beta }{\beta (1-\xi )}\right)$$
and
(23)$$\psi \left(\xi \right)=\frac{\alpha +\xi \beta }{\sqrt{2\pi \beta (1-\beta )(1-\xi )(1-\xi \beta )}}.$$The partition function, thus, takes the form pertinent to the application of the Laplace method. It is easy to see that the stationary point $\xi_{0}$ defined by $\phi^{\prime }\left(\xi \right)=0$ is given by
(24)$$\xi_{0}=\frac{\beta p-1}{\beta p-\beta }$$
and belongs to interval $0\leq \xi_{0}\leq 1$ if β∈[1/p,1] and, thus, α∈[0,(p−2)/p], while for larger α, the maximal value of ϕ(ξ) is reached for ξ=0. Thus, the Laplace approximation gives the leading asymptotic of $Z_{t}\left(\alpha t\right)$ for any fixed α. If α<(p−2)/p it gives
(25)$$Z_{t}\left(\alpha t\right)\approx {\left(p-1\right)}^{t(1-\beta )}\beta p\sqrt{t}\sqrt{\frac{-2\pi }{t\phi^{\prime \prime }\left(\xi_{0}\right)}}\psi \left(\xi_{0}\right)e^{t\phi (\xi_{0})}=p\frac{p-2}{p-1}p^{t},$$
and $Z_{t}\left(\alpha t\right)=Z_{0}\left(t\right)$ is an α-independent constant, while for α>(p−2)/p
(26)$$\begin{array}{c} Z_{t}\left(\alpha t\right)\approx -{\left(p-1\right)}^{t(1-\beta )}\beta p\sqrt{t}\frac{\psi \left(0\right)e^{t\phi (0)}}{t\phi^{\prime }\left(0\right)}=\\ =\frac{-\alpha p}{\ln\left(p\beta \right)\sqrt{2\pi \beta (1-\beta )t}}{\left[\beta^{-\beta }{\left(\frac{p-1}{1-\beta }\right)}^{\left(1-\beta \right)}\right]}^{t}\ll Z_{0}\left(t\right). \end{array}$$Thus, in the limit of large *t*, the solution has the shape of a traveling wave. The probability distribution of the position of the trajectory’s endpoint
(27)$$P\left(x,t\right)=\frac{Z_{t}\left(x\right)}{\sum_{y}Z_{t}\left(y\right)}$$
is constant up to x=vt, where v=(p−2)/p, and is asymptotically zero for a larger *x*. To elucidate the shape of the wave near x=vt, note that the Laplace approximation implies replacing the limits of integration in (Equation 21) with plus and minus infinity. For a better approximation of $x=vt+\Delta \sqrt{t}$, one should replace the integrand in (Equation 21) with the corresponding Gaussian function, but retain the correct integration limit, giving
(28)$$\begin{array}{c} Z_{t}\left(vt+\Delta \sqrt{t}\right)\approx {\left(p-1\right)}^{t(p-1)/p}\sqrt{t}\psi \left(\xi_{0}\right)e^{t\phi (\xi_{0})}\int_{0}^{\infty }e^{t\frac{{\left(\xi -\xi_{0}\right)}^{2}}{2}\phi^{\prime \prime }\left(\xi_{0}\right)}d\xi =\\ =\frac{1}{2}p\frac{p-2}{p-1}p^{t}\mathrm{erfc}\left(\frac{p\Delta }{\sqrt{8(p-1)}}\right). \end{array}$$Figure 3 shows that this expression provides a satisfactory approximation to the shape of the wave at Δ.

## 4. Critical Behavior on Finite Graphs

Thus, we have proven in the previous section that the distribution of the endpoint of a *t*-step trajectory on an infinite critical tree reaches a limiting shape of the form $\mathrm{erfc}\left[c\left(x-vt\right)/\sqrt{t}\right]$, moving toward infinity with constant velocity v=(p−2)/p. In a finite system, this process cannot continue forever: at some point, the expanding wave feels that the system is finite. Clearly, the number of steps at which this occurs is determined by the following:(29)$$\frac{\ln N}{\ln(p-1)}\approx vt\;\;\to \;t\approx t_{0}=\frac{p}{p-2}\frac{\ln N}{\ln(p-1)}$$
where *N* is the total number of nodes in the graph. Further evolution depends on how exactly the finite tree is defined. Indeed, the notion of a “finite tree with constant degree p>2” is ill-defined. A finite tree of *N* nodes has exactly N−1 bonds and, therefore, must have an average degree 2(N−1)/N≈2. This discrepancy is usually resolved in one of two possible ways. One option is to consider a *k*-generation tree; it has a boundary layer consisting of $p_{0}{\left(p-1\right)}^{k-1}$ nodes with degree 1. These nodes maintain a final fraction of the system size when k→∞ and shift the average node degree to 2. Another option is to consider a graph in which all nodes have degree *p* (except for the special node with degree $p_{0}$), but which is only locally tree-like. This means that while in such a graph there are no short cycles (i.e., it is locally tree-like), it does contain long cycles with a minimum length on the order of $t_{0}$. We will call this option a random regular graph (RRG). A more detailed discussion of the terminology is presented in the next section.

The qualitative trajectory behaviors in both cases are similar to the behaviors on an infinite tree only up to a trajectory length of around $t_{0}$. For larger trajectories, the behavior is radically different.

Indeed, on a large but finite tree, whenever the expanding distribution reaches the boundary, it becomes stationary; for any $t>t_{0}$, it remains essentially the same, namely, the probability of the trajectory to the end at distance *x* from the root is *x*-independent and roughly (up to some minor deviations near x=0 and x=k) equals:(30)$$P\left(x,t\right)=\left\{\begin{array}{cc} 2/k & \;\;\mathrm{if}\;t+x\;\mathrm{is}\;\mathrm{even},\\ 0 & \;\;\mathrm{if}\;t+x\;\mathrm{is}\;\mathrm{odd} \end{array}\right.$$
which means that the probability of ending the trajectory at any given node of the tree is, in fact, proportional to ${\left(p-1\right)}^{-x}$ (considering that there are roughly $p{\left(p-1\right)}^{x}$ nodes at a distance *x* from the root).

However, as we show in the next section, both the limiting behavior and the typical length needed to reach it, are radically different in the case of the RRG. Indeed, in the limit of large *t*, the trajectory’s endpoint distribution consists of two zones, namely, the zone close to the root, where the likelihood of finding the trajectory’s end decays with *x* at a rate of ${\left(p-1\right)}^{-x}$, and the remaining bulk of the system, where the probability of finding the trajectory end remains position-independent. However, the trajectory length $t_{relax}$ needed to reach this final state is extremely long, $t_{relax}∼\sqrt{N}$ (compare this with $t_{0}∼\ln N$).

### Random Regular Graph (RRG)

Consider a graph constructed as follows. Take a *k*-generation tree with degree *p* at all branching points except the root, and degree $p_{0}=p_{cr}=p\left(p-1\right)$ in the root. This tree has $p{\left(p-1\right)}^{k}$ “leaves” (periphery nodes of degree one). Now, add $p{\left(p-1\right)}^{k+1}/2$ new bonds connecting the leaves in a way that each leave becomes a node of degree *p*. Such graphs are reminiscent of the random regular graph (RRG) ensemble (which is defined as a maximal entropy ensemble of graphs whose nodes have a constant degree *p*) in the sense that it is locally tree-like (the probability of forming a short cycle is exponentially small in *k*), and all nodes, except for the root, have a constant degree *p*. The most important difference between the graph defined above and the true RRG ensemble is as follows: in the former the shortest cycle, which includes the root, has, by construction, length 2k, while in the latter there exist shorter cycles, which include the root. However, the fraction of such cycles exponentially decreases with 2k−l; therefore, we expect the difference to be negligible.

To check whether the resulting graphs are similar to the true RRG, we calculate (see Figure 4) the eigenvalue spectrum of an ensemble of such graphs (trees with randomly connected external nodes) for $p_{0}=p$ and compare it to the known limiting distribution for the true RRG ensemble [19,20]
(31)$$\rho \left(\lambda \right)=\left\{\begin{array}{cc} \frac{p\sqrt{4\left(p-1\right)-\lambda^{2}}}{2\pi (p^{2}-\lambda^{2})} & \mathrm{for}\;|\lambda |<2\sqrt{p-1},\\ 0 & \mathrm{otherwise}. \end{array}\right.$$Clearly, the distribution is very close to the limiting one, except for *k* delta-functional peaks reflecting the existence of a regular tree sub-graph. However, the relative sizes of these peaks decrease, and for a large *k*, the distribution converges to the Kesten–McKay result (Equation 31). This result is, in fact, to be expected, given that the only assumptions needed to prove (Equation 31) (see [20]) are (i) that all nodes have the same degree *p*, and (ii) the concentration of a cycle of any finite length *m* converges to zero with the growing size of the network; both are true for the suggested tree closure construction.

For the purposes of this paper, the most interesting question is the behavior at the edge of the spectrum, where the properties of the true RRG ensemble, and our model RRGs are exactly the same. For $p_{0}=p$, there is a single maximal eigenvalue $\lambda_{1}=p$, reflecting the fact that—on each next step—every trajectory has exactly *p* possible continuations, and the continuous spectrum with density converges to 0 as $\rho \left(\lambda \right)∼\sqrt{\lambda_{2}^{2}-\lambda^{2}}$, when $\left|\lambda \right|\to \lambda_{2}=2\sqrt{p-1}<\lambda_{1}$, so there is a *k*-independent final gap between the maximal and second-largest eigenvalue.

Let us now calculate the distribution of the endpoints of a *t*-step trajectory on such a quasi-RRG graph. To do that, we introduce a (k+1)-dimensional vector $Z_{t}$, so that the *x*-th component of this vector equals the number of *t*-step trajectories starting from the root and ending at distance *x*.

The recurrent relation for the number of paths can be written in matrix form:(32)$$Z_{t+1}=TZ_{t}\;\;\;\;\;\;\;Z_{0}={\left(1,0,0,\ldots \right)}^{T}$$
with the tri-diagonal (k+1)×(k+1) transfer matrix **T**
(33)$$T=\left(\begin{array}{cccccc} 0 & 1 & 0 & 0 & 0 & \ldots \\ p_{0} & 0 & 1 & 0 & 0 & \ldots \\ 0 & p-1 & 0 & 1 & 0 & \ldots \\ \ldots & \ddots & \ddots & \ddots & \ddots & \ldots \\ \ldots & 0 & 0 & p-1 & 0 & 1\\ \ldots & 0 & 0 & 0 & p-1 & p-1 \end{array}\right).$$Note the (p−1) entry in the lower-left corner of the transfer matrix. It corresponds to the trajectories “jumping” from one node of the *k*-th generation to the other along the random connection bonds shown in Figure 5. It is this single entry that distinguishes the transfer matrix of the RRG graph problem from the transfer matrix $T_{0}$ of the finite tree one, where this entry equals 0.

It is easy to see that all eigenvalues of both T and $T_{0}$ are non-degenerate and real. Indeed, it is sufficient to note that $\Omega^{-1}T\Omega$, where Ω is a diagonal matrix with elements
(34)$$\Omega_{00}=\sqrt{\frac{p-1}{p_{0}}},\;\;\;\Omega_{xx}={\left(p-1\right)}^{\frac{x}{2}}\;\mathrm{for}\;1\leq x\leq k,$$
is a symmetric tri-diagonal matrix. The decomposition of T allows us to write
(35)$$Z_{t}=\sum_{i}\lambda_{i}^{t}\left|i\rangle \langle i\right|Z_{0}$$
where index *i* enumerates the eigenvalues and eigenvectors of T. Assume for convenience that the eigenvalues are enumerated in decreasing order of their absolute value: $|\lambda_{1}\left|>\right|\lambda_{2}\left|>\right|\lambda_{3}|>\ldots$. Clearly, the large *t*, asymptotic of the partition function, is controlled by the eigenvalues with the largest absolute values and corresponding eigenvectors.

The eigenvalue problem for the transfer matrix $T_{0}$ has been studied at length in [8,21]. The characteristic polynomials of that problem $P_{k}\left(\lambda \right)$ satisfy
(36)$$\left\{\begin{array}{ccc} P_{-1} & = & \frac{p_{0}}{p-1},\\ P_{0} & = & \lambda ,\\ P_{k} & = & \lambda P_{k-1}-\left(p-1\right)P_{k-2},\;\;\;k\geq 1 \end{array}\right.$$And have the explicit form
(37)$$P_{k}=C_{+}\mu_{+}^{k+1}+C_{-}\mu_{-}^{k+1},$$
where
(38)$$\mu_{\pm }=\frac{\lambda \pm \sqrt{\lambda^{2}-4\left(p-1\right)}}{2}=\sqrt{p-1}u^{\pm 1}$$
and
(39)$$C_{\pm }=\frac{p_{0}}{2(p-1)}\mp \frac{\lambda (p_{0}-2\left(p-1\right))}{2\left(p-1\right)\sqrt{\lambda^{2}-4\left(p-1\right)}}=\frac{p_{0}}{2(p-1)}\mp \left(\frac{p_{0}}{2(p-1)}-1\right)\frac{u^{2}+1}{u^{2}-1},$$
where we introduce parametrization
(40)$$\lambda =\sqrt{p-1}\left(u+1/u\right).$$Notably, parametrization (Equation 40) means that *u* is real for $\left|\lambda \right|\geq 2\sqrt{p-1}$ and imaginary otherwise.

To proceed further, replace $p_{0}=p\left(p-1\right)$ and note that characteristic polynomials D(λ)=det(λI−T) of the RRG transfer matrix T are easily expressed in terms of $P_{k}\left(\lambda \right)$
(41)$$D_{k}=\left(\lambda -p+1\right)P_{k-1}-\left(p-1\right)P_{k-2}.$$Given that we are interested in eigenvalues with the largest absolute value, we concentrate here on the real solutions of the characteristic equation $D_{k}\left(u\right)=0$. After substituting (Equation 37)–(Equation 40) into (Equation 41), we rewrite it in the form:(42)$$u^{-2(k+2)}=\frac{\left(1+v/u\right){\left(1-v/u\right)}^{2}}{\left(1+vu\right){\left(1-vu\right)}^{2}}$$
where we introduce notation $v=\sqrt{p-1}$. For every *k*, this equation has three roots, $u_{1,2,3}$, converging to the zeros of the numerator for the large *k*:(43)$$u_{1,3}=v+\epsilon_{1,3},\;\;u_{2}=-v+\epsilon_{2},\;\;\epsilon_{1,2,3}\to 0\;\mathrm{for}\;k\to \infty$$Expanding (Equation 42) up to second order in ϵ and solving the resulting equations, we obtain the following finite-size correction terms for positive
(44)$$\begin{array}{c} \epsilon_{1,3}=\pm \frac{v^{2}-1}{v}\sqrt{\frac{\left(1+v^{2}\right)}{2}}v^{-k}-\frac{\left(1+v^{2}\right){\left(v^{2}-1\right)}^{2}}{2v^{3}}kv^{-2k}+\\ +\frac{\left(v^{2}-1\right)\left(3v^{4}+2v^{2}+3\right)}{8v^{3}}v^{-2k}+o\left(v^{-2k}\right) \end{array}$$
and negative
(45)$$\epsilon_{2}=\frac{\left(v^{2}-1\right){\left(1+v^{2}\right)}^{2}}{4v^{3}}v^{-2k}+o\left(v^{-2k}\right)$$
eigenvalues, respectively. After returning to the original variables, we obtain
(46)$$\begin{array}{c} \lambda_{1,3}=p\pm \sqrt{\frac{p}{2}}\frac{{\left(p-2\right)}^{2}}{p-1}{\left(p-1\right)}^{-k/2}-\frac{p{\left(p-2\right)}^{3}}{2{\left(p-1\right)}^{2}}k{\left(p-1\right)}^{-k}+\\ +\frac{{\left(p-2\right)}^{2}\left(3p^{2}+4\right)}{8{\left(p-1\right)}^{2}}{\left(p-1\right)}^{-k}+o\left({\left(p-1\right)}^{-k}\right) \end{array}$$
and
(47)$$\lambda_{2}=-p+\frac{p^{2}{\left(p-2\right)}^{2}}{4{\left(p-1\right)}^{2}}{\left(p-1\right)}^{-k}+o\left({\left(p-1\right)}^{-k}\right).$$Figure 6a,b show the asymptotic behaviors of the discrepancies $\lambda_{i}-\lambda_{i}\left(k=\infty \right)$ as compared to their approximate values given by (Equation 46) and (Equation 47).

We notice that, in the limit of the large number of generations, the gaps between the absolute values of the three leading eigenvalues are exponentially small:(48)$$\Delta \lambda_{1}\approx \Delta \lambda_{2}\approx \sqrt{\frac{p}{2}}\frac{{\left(p-2\right)}^{2}}{p-1}{\left(p-1\right)}^{-k/2},$$
where we use notation $\Delta \lambda_{i}=|\lambda_{i}\left|-\right|\lambda_{i+1}|$, indicating that trajectories on the order of ${\left(p-1\right)}^{k/2}∼\sqrt{N}$ steps are needed to reach the asymptotic distribution of the endpoints. Note that since there are only three real solutions of (Equation 42), the remaining eigenvalues
(49)$$|\lambda_{i}|\leq 2\sqrt{p-1},\;\;i\geq 4$$
and these are separated from the triplet defined by (Equation 46) and (Equation 47).

To understand the limiting distribution of the endpoints, as well as the modes that demonstrate a remarkably slow decay, let us find the eigenvectors corresponding with these three leading eigenvalues. Let $\Psi_{i}$ be the *i*-th eigenvector, i.e., a solution to
(50)$$\left(\lambda_{i}I-T\right)\Psi_{i}=0,$$
and $\Psi_{x}^{\left(i\right)}$ be its *x*-th element. Choosing the initial condition $\Psi_{0}^{\left(i\right)}=1$ and solving recursively with respect to *x*, we obtain
(51)$$\Psi_{x}^{\left(i\right)}=P_{x-1}\left(\lambda_{i}\right)\;\;\;\;x=1,\ldots ,k,$$
where $P_{x-1}\left(\lambda \right)$ is defined above in (Equation 37). Noticing that
(52)$$P_{x-1}\left(\lambda =p\right)=p;\;\;\;P_{x-1}\left(\lambda =-p\right)={\left(-1\right)}^{x}p$$
and expanding $P_{x-1}\left(\lambda \right)$ in the vicinity of these points, we obtain the following in the leading order:(53)$$\Psi_{x}^{\left(1,3\right)}\approx p\pm \sqrt{2p}{\left(p-1\right)}^{\left(x-\frac{k}{2}\right)}$$
and
(54)$$\Psi_{x}^{\left(2\right)}\approx {\left(-1\right)}^{x}\left(p+\frac{p}{2(p-1)}{\left(p-1\right)}^{\left(x-k\right)}\right).$$Thus, from the point of view of the senior eigenvector $\Psi_{1}$, the graph consists of two different zones: $x≲\frac{k}{2}$ and $x≳\frac{k}{2}$. In the first zone (see Figure 7), the probability distribution over *x* is flat, while in the second, it grows as ${\left(p-1\right)}^{x}$, i.e., proportionally to the number of nodes in the *x*-generation. Thus, the probability of the trajectory ending at a *given node* is position-independent in the second zone and proportional to ${\left(p-1\right)}^{-x}$ (where *x* is the distance from the node to the route) in the first zone.

Turning to the distribution of the trajectory endpoint:(55)$$P\left(x,t\right)=\frac{Z_{t}\left(x\right)}{\sum_{y}Z_{t}\left(y\right)}$$
for the finite trajectory length, and combining the results with what we know from the previous section, we understand that there are two very distinct regimes in the evolution of this probability distribution. For small $t≲t_{0}$, where $t_{0}$ is defined by (Equation 29), the trajectory is yet to reach the boundary, so the probability has the shape of an expanding wave studied in the previous section (Equation 28). At around $t_{0}$, the wave reaches all points of the graph and the distribution becomes numerically very close to (Equation 30):(56)$$P\left(x,t_{0}\right)\approx \left(1+{\left(-1\right)}^{t_{0}+x}\right)\frac{1}{k}\approx \frac{1}{N}\left(\Psi_{x}^{\left(1\right)}+\Psi_{x}^{\left(3\right)}+2{\left(-1\right)}^{t_{0}}\Psi_{x}^{\left(2\right)}\right),$$
where the last equation provides the approximate linear expansion of $P(x,t_{0})$ in terms of the leading eigenvectors, and N is a normalizing coefficient. Thus, for $t>t_{0}$, one expects
(57)$$\begin{array}{ccc} P(x,t) & ∼ & \lambda_{1}^{t-t_{0}}\Psi_{x}^{\left(1\right)}+\lambda_{3}^{t-t_{0}}\Psi_{x}^{\left(3\right)}+2{\left(-1\right)}^{t_{0}}\lambda_{2}^{t-t_{0}}\Psi_{x}^{\left(2\right)}\\ \; & ∼ & \Psi_{x}^{\left(1\right)}+{\left(1-\frac{2\Delta \lambda_{1}}{\lambda_{1}}\right)}^{t-t_{0}}\Psi_{x}^{\left(3\right)}+2{\left(-1\right)}^{t}{\left(1-\frac{\Delta \lambda_{1}}{\lambda_{1}}\right)}^{t-t_{0}}\Psi_{x}^{\left(2\right)}, \end{array}$$
so it takes approximately
(58)$$t_{relax}=\frac{\lambda_{1}}{\Delta \lambda_{1}}=\sqrt{2p}\frac{p-1}{{\left(p-2\right)}^{2}}{\left(p-1\right)}^{k/2}\approx \frac{\sqrt{2(p-1)}}{{\left(p-2\right)}^{3/2}}\sqrt{N}$$
steps to reach the asymptotic limit (here,
(59)$$N=1+\frac{p(p-1)}{p-2}\left[{\left(p-1\right)}^{k}-1\right]$$
is the total number of nodes in the graph).

## 5. Discussion

In this paper, we studied the behavior of the path-counting partition function on the infinite tree, and on a random regular graph with a point-like disorder in the case of critical disorder strength. We show that, while on an infinite tree the evolution of the probability distribution at the end of the path has a step function form that expands with the constant speed v=(p−2)/p, as conjectured in [8], the behavior on the finite RRG is much richer and consists of two regimes. There is a fast (on the order of $t_{0}∼\log_{p-1}N$) expanding wave regime, similar to the evolution of the infinite system, followed by a much slower relaxation with maximal relaxation time (on the order of $t_{relax}∼\sqrt{N}$). This regime is an example of a *critical slowdown* and, remarkably, it is completely unattainable when studying infinite systems. Indeed, while for any reasonable value of *N*$t_{0}$ may seem quite small, it formally diverges as N→∞.

The existence of the critical slowdown regime can be qualitatively understood from the fact that, in the thermodynamic limit, at the critical point, the transfer matrix of the problem has multiple eigenvalues with exactly the same absolute value (in the case considered here, there are three of them, reflecting a conservation of the parity of t+x, in the absence of such additional conservation laws, there are typically two). Meanwhile, the maximal eigenvalue of the adjacency matrix of a finite connected graph is non-degenerate [1], which dictates that there must be a finite gap between these eigenvalues in any finite system. It is natural to expect this gap to be quite small in the case of a large graph, that is, to be much smaller than what one expects away from the critical point where the gap remains finite even in the infinite graph. This qualitative consideration explains why one expects the critical slowdown to occur.

One expects that such a slowdown is a general phenomenon, which should be typical in the study of localization transitions on other large but finite graphs. One example is the Anderson localization on the Bethe lattice [22,23] (also see review [24]) and on RRG [25,26], which has been actively studied recently due to its suggested connection to the many-body localization problem [27]. Indeed, the exponential divergence of the diffusion constant in the vicinity of the critical point [22] is reminiscent of the critical slowdown discussed here.

Another relevant example is the celebrated quasispecies model [28] of biological evolution. The model can be briefly formulated as follows. Consider a set of possible genomes $œ=(\sigma_{1},\sigma_{2},\ldots ,\sigma_{d})$, consisting of *d* binary variables $\sigma_{i}=0,1$, so that there are $2^{d}$ possible genomes. We introduce a population function Z(œ,t), reflecting the occurrence of each genome at time *t*. Then a discrete time evolution of the quasispecies model in the absence of the niche size constraint reads
(60)$$Z\left(œ,t+1\right)=\alpha_{œ}\sum_{{œ}^{\prime }}\Pi_{œ,{œ}^{\prime }}Z\left({œ}^{\prime },t\right)$$
where $\alpha_{œ}$ is the fitness of the genome œ and $\Pi_{œ,{œ}^{\prime }}$ is the probability of obtaining œ while copying ${œ}^{\prime }$, allowing for possible copying errors. In this system, depending on the probability of copying errors, a localization–delocalization transition, often called the error threshold or error catastrophe [28,29] (also see the pedagogical presentation in [30]) is possible. In the localized state, the observed genomes are close to the optimal one (i.e., the one with the highest fitness) and form a localized cloud known as a quasispecies, while in the delocalized state, all genomes appear with equal probability.

We make numerical simulations of this system for the simplest case, where
(61)$$\alpha_{œ}=1+\left(\alpha -1\right)\delta_{œ,{œ}_{0}},\;\;\;\Pi_{œ,{œ}^{\prime }}=\left(1-q\right)\delta_{d_{H},0}+\frac{q}{d}\delta_{d_{H},1}$$
where ${œ}_{0}$ is the genome with optimal fitness, α>0 and q∈(0,1) are parameters, and $d_{H}=d_{H}\left(œ,{œ}^{\prime }\right)$ is the Hamming distance (the number of differences in genome letters) between genomes œ and ${œ}^{\prime }$. For simulations, we use d=60 and q=0.5 and find that localization transition occurs at α=d−1. Moreover, Figure 8 shows that there is strong evidence of critical slowdown at the critical point (Figure 8b), where the number of steps of the time evolution needed to approach the limiting distribution is 1-2 orders of magnitude larger than in the bulk of localized (Figure 8a) and delocalized (Figure 8c) states. We are not able to solve this model analytically but we suggest that the simpler, exactly solvable model presented in this paper can serve as a toy model for this more complex one.

## Figures and Tables

**Figure 1 entropy-25-01318-f001:**
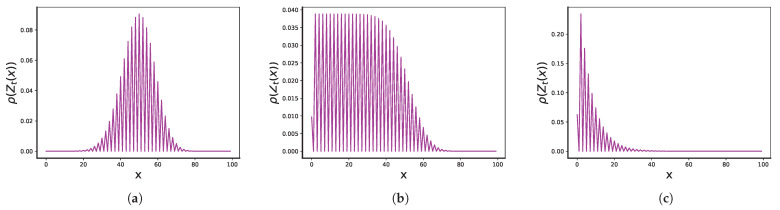
Distribution P(x,t) for trajectories of t=100 steps for three different combinations of *p* and $p_{0}$: (**a**) delocalized regime, $p=4,p_{0}=7$, (**b**) critical regime $p=4,p_{0}=12$, and (**c**) localized regime, $p=4,p_{0}=15$.

**Figure 2 entropy-25-01318-f002:**
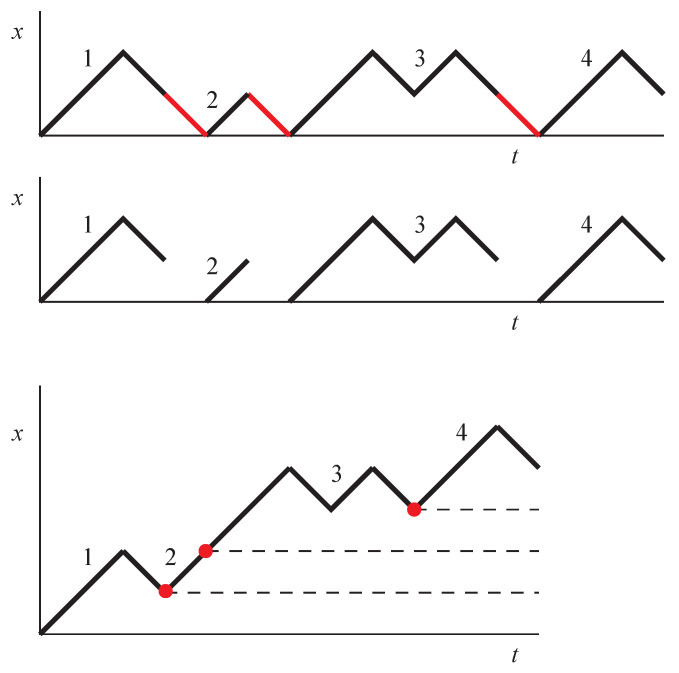
Illustration of the bijection between trajectories from (0,0) to (t,x) visiting 0 *m* times and trajectories from (0,0) to (t−m,x+m) not visiting zero, reflected in Equation (Equation 16). To get from trajectory shown in the top panel to that shown in the bottom panel, remove the steps highlighted in red, and move the remaining pieces so they are joined together. To get from trajectory shown in the bottom panel to that shown in the top panel, insert (1,−1) steps (red lines) in place of the red dots. Red dots are points where x=1,2,…,m for the final time, i.e., rightmost intersections of the trajectory and x=1,2,…,m lines (dotted lines).

**Figure 3 entropy-25-01318-f003:**
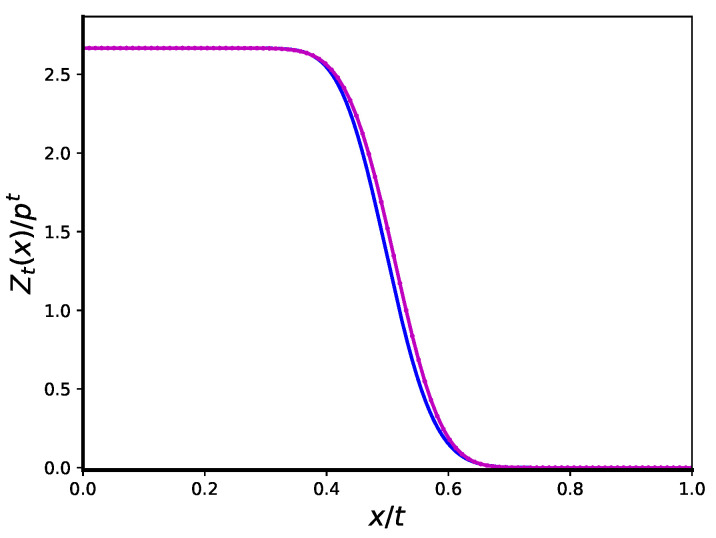
Numerical (magenta) and asymptotic (blue) envelopes of the wave $Z_{t}\left(\alpha t\right)p^{-t}$ as functions of α for p=4, $p_{0}=p_{cr}=p\left(p-1\right)=12$, t=200.

**Figure 4 entropy-25-01318-f004:**
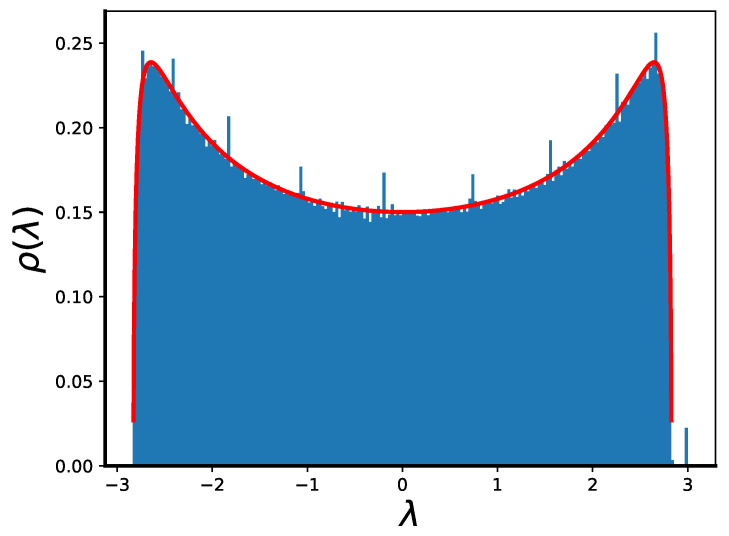
Eigenvalue spectrum of the approximate RRG graph created by a procedure shown in Figure 5 for p=3,k=9 averaged over 100 independent realizations (histogram) vs. the limiting spectrum of the true large RRG graph given by (Equation 31) (red line). Note the spectral gap between the largest eigenvalue $\lambda_{1}=p=3$ and the edge of the continuous spectrum at $\lambda =2\sqrt{p-1}\approx 2.828$.

**Figure 5 entropy-25-01318-f005:**
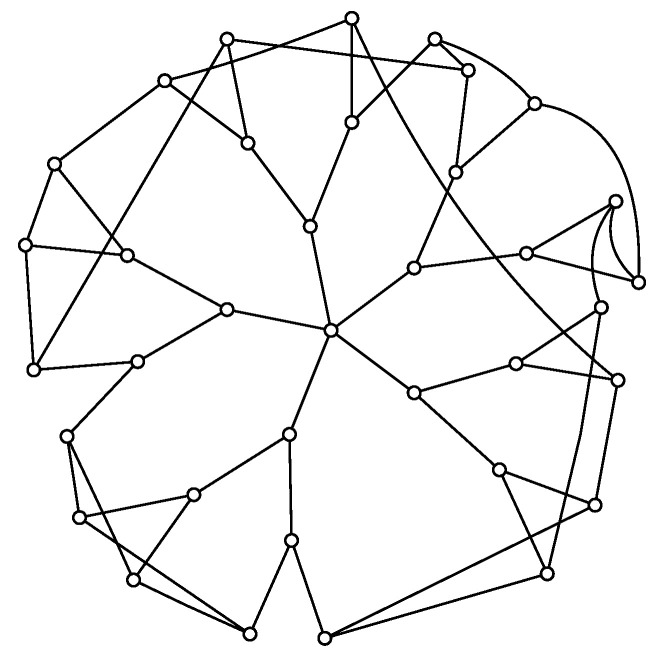
A sketch of a model graph with a fixed degree p=3, except for the root; up until the k=3’rd generation from the root with degree $p_{0}=5$, the graph is a tree, and the outer nodes are connected randomly with each other so that their degree equals *p*.

**Figure 6 entropy-25-01318-f006:**
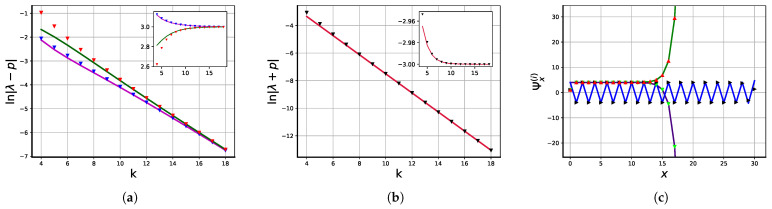
(**a**,**b**): Largest eigenvalues of the transfer matrix T as functions of *k* for p=3 in the logarithmic (main) and linear (inset) scales. (**a**): Absolute value of the difference between positive eigenvalues $\lambda_{1,3}$ and *p*; numerical results are shown in blue ($\lambda_{1}$) and red ($\lambda_{3}$) triangles; asymptotic (Equation 46) is shown by magenta and green lines. (**b**) Absolute value of the difference between the negative eigenvalue $\lambda_{2}$ and −p with an inserted plot of eigenvalues in a normal scale. (**c**): Components of eigenvectors $\Psi^{\left(i\right)},i=1,2,3$ as functions of coordinate *x* for p=4,k=30; numerical values and asymptotic given by (Equation 53) and (Equation 54) are shown. $\Psi^{\left(1\right)}$: Red triangles and green line; $\Psi^{\left(2\right)}$: black triangles and blue line; $\Psi^{\left(3\right)}$: light green triangles and purple line, respectively.

**Figure 7 entropy-25-01318-f007:**
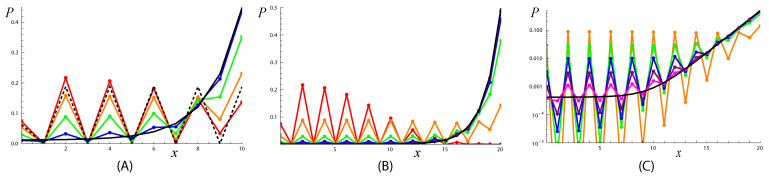
Evolution of the end-of-trajectory probability distribution $P\left(x,t\right)=Z_{t}\left(x\right){\left(\sum_{y}Z_{t}\left(y\right)\right)}^{-1}$ on an RRG with (**A**) k=10 and (**B**,**C**) k=20 generations and branching degree p=3. The branching degree of the root is critical, $p_{0}=p_{cr}=p\left(p-1\right)=6$. The limiting distribution $P_{\infty }\left(x\right)=\Psi_{x}^{\left(1\right)}{\left(\sum_{y}\Psi_{y}^{\left(1\right)}\right)}^{-1}$, (Equation 53) is shown by the black line. (**A**) The results for trajectories with lengths t=20 (red, propagation regime), 30 (orange, crossover from the propagation to slow relaxation regime), 60 (green), 200 (blue). The intermediate asymptotic (Equation 56) is shown by the dashed line; (**B**) the results for trajectories with lengths t=20 (red, propagation regime), 60 (orange, crossover from propagation to slow relaxation regime), 200 (green), 600 (blue), 2000 (purple), 6000 (magenta); the intermediate asymptotic (Equation 56) is almost indistinguishable from the orange line; (**C**) the same as (**B**) but in logarithmic coordinates.

**Figure 8 entropy-25-01318-f008:**
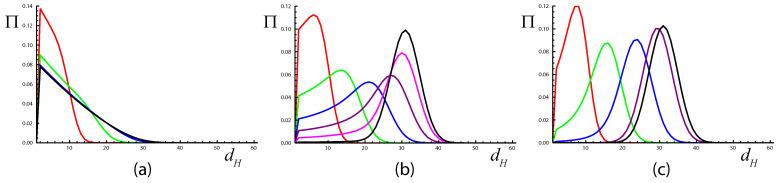
Probability Π of observing a genome at a distance $d_{H}$ from ${œ}_{0}$ after *t* steps of evolution (Equation 60) and (Equation 61) starting from the initial condition $Z\left(œ,0\right)=\delta_{œ,{œ}_{0}}$ for (**a**) localized, (**b**) critical, and (**c**) delocalized states of the quasispecies model. Parameters are d=60,q=0.5,α= (**a**) 69, (**b**) 59, and (**c**) 49. In all panels, black lines represent the limiting distribution (the eigenvector corresponding to the largest eigenvalue of the transfer matrix), colored lines represent the distributions after t=20 (red), 60 (green), 200 (blue), 600 (purple), and 2000 (magenta) steps of the evolution. The distribution $\Pi (d_{H},t)$ is defined in a natural way: $\Pi \left(d_{H},t\right)=\sum_{œ}Z\left(œ,t\right)\delta_{d_{H}\left(œ,{œ}_{0}\right),d_{H}}/\sum_{œ}Z\left(œ,t\right)$.

## Data Availability

Not applicable.

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
