# Peer review of "Path Counting on Tree-like Graphs with a Single Entropic Trap: Critical Behavior and Finite Size Effects"

_entropy, 2023, doi:10.3390/e25091318_

Round 1
Reviewer 1 Report
The papr provides a very good piece of work devoted to the behavior of random paths on trees with an entropic trap at the origin. It is known that with increasing the depth of a tap the paths get locallized at the tree root. The authors consider the paths' statistics exactly at the localization transition point and found a number of interesting features.
I recommend this paper for publication, however I would like to pay attention to some points which deserve clarification before acceptance of the paper.
1) The notion of a "random walk" implies that there are probabilities of local transitions on each step with a total conservtion law. However the authors deal not with the probability, but with the partition fuction for which they write recursion relations (3) and (8). The phase transition emerges just is a paths counting problem, but not in a "random walk" problem with conserved probabilities (in the last case there is no any entropic trap in a system). So, to not to confuse a reader I would suggest to speak not about random walks, but about paths counting.
2) I have some doubts concerning a "RRG part" of the paper. The authors have replaced the true RRG by an effective tree with a closure at the boundary (see the transfer operator (28)). I am not sure that all features of real RRG graphs are reproduced correctly. The computation of a spectral density for such an effective graph is highly be desirable: matching the Kesten-Mckay law would be rather convincing proof of validity of such a consideration.
In all other aspects I find the work original and interesting. I can recommend it for publication in Entropy.
Everything is almost OK except some misspellings. Some sentences could be simplified for better understanding.
Author Response
We are very grateful to both referees and to the scientific editor for their complimentary assessment of our work and for their interesting and insightful comments. We incorporated the discussion of the points raised by the referees into the text, which, at least in our opinion, significantly improved its quality.
Below we address the comments of the referees one by one, outlining the changes made in the manuscript to address them:
Comments of the first referee:
1) The notion of a "random walk" implies that there are probabilities of local transitions on each step with a total conservation law. However the authors deal not with the probability, but with the partition function for which they write recursion relations (3) and (8). The phase transition emerges just is a paths counting problem, but not in a "random walk" problem with conserved probabilities (in the last case there is no any entropic trap in a system). So, to not to confuse a reader I would suggest to speak not about random walks, but about paths counting.
We are grateful to the referee for this comment, which, together with the first comment of referee 2 allowed us to better understand the difference between the "path counting" and "maximal entropy random walk" problems and improve the presentation of this question. In fact, the referee is partly right: it seems indeed impossible to create a random walk process, which generates distributions of the type (4) (i.e., equiprobable distribution over all trajectories of the same length) on each step. But it is possible to construct a random walk (i.e., a Markovian process with conservation of probability on each step) solving a somewhat easier problem, that is, generating a measure, which is equiprobable over all trajectories of the same length {\it with the same start and end points}. And this is exactly what is called a "maximal entropy random walk".
Notably, if one considered a walk on a finite graph, and considered the limit of infinite number of steps, the influence of the ends of trajectory becomes negligible, and distribution of the inner links of this walk becomes described by the square of the dominant eigenvector of the adjacency matrix in a way similar to the density of polymer links in the "ground state dominance" regime for an ideal polymer in external field. We added three paragraphs discussing this topic to the introduction.
Reviewer 2 Report
The authors discuss the problem of path counting in Bethe lattice (i.e. infinite Cayley tree). They are interested in the paths initiated at the root of the tree, and in particular the probability of finding the end of the path at distance x in time t. From the previous work (reference [7]) it was known that the probability changes the behavior from delocalized to located at the critical root degree p_cr = p(p-1), where p is the degree of vertices other than the root.
The main result of this work is the proof of the traveling wave formula postulated in [7] for this probability in the critical case.
The rest of the paper is devoted to the analysis of finite graphs that are identical to the Bethe lattice at a distance n or less from the root.
Such graphs are constructed as n-generation Cayley trees, whose leaves are connected by (p-1) new edges to other leaves in a random manner. The path counting problem in such graphs is a finite version of the original problem for Bethe networks: Infinite recursive equations (8) are stopped at the nth generation. Paths from the nth generation do not go to the (n+1)th generation (which does not exist), but remain within the nth generation. The authors discuss the finite size-effects of this particular cut-off graphs.
I recommend the paper for publication. However, I would like to draw the authors' attention to the following points:
(1)
First of all, the path counting problem is related but not identical to the maximal entropy random walk. The main difference is that in addition to the number of paths the formula for the probability of finding the end-point of the path at some position includes the ratio of the eigenvector components for the initial and last nodes of the eigenvector associated with the leading eigenvalue of the adjacency matrix. It also includes the t-th power of this eigenvalue. This makes enormous difference,
(see Z. Burda, J. Duda, J.M. Luck, B. Waclaw, Acta Phys. Polon. B 41, 949 (2010) https://arxiv.org/abs/1004.3667 for discussion (in particular section 4))
(2)
Second of all, the maximal entropy random walk on Cayley tree was studied analytically in
Z. Burda and J. Ochab, Phys. Rev. E 85, 021145 (2012), https://arxiv.org/abs/1201.1420
What is interesting there is no sign of the critical value p_cr = p(p-1), instead the solution changes behaviour at p_cr = 2p.
Author Response
We are very grateful to both referees and to the scientific editor for their complimentary assessment of our work and for their interesting and insightful comments. We incorporated the discussion of the points raised by the referees into the text, which, at least in our opinion, significantly improved its quality.
Below we address the comments of the referees one by one, outlining the changes made in the manuscript to address them:
Comments of the second referee
(1)
First of all, the path counting problem is related but not identical to the maximal entropy random walk. The main difference is that in addition to the number of paths the formula for the probability of finding the end-point of the path at some position includes the ratio of the eigenvector components for the initial and last nodes of the eigenvector associated with the leading eigenvalue of the adjacency matrix. It also includes the t-th power of this eigenvalue. This makes enormous difference,
(see Z. Burda, J. Duda, J.M. Luck, B. Waclaw, Acta Phys. Polon. B 41, 949 (2010) https://arxiv.org/abs/1004.3667 for discussion (in particular section 4))
We are grateful to the referee for pointing out this. We rewrote the introduction to make the difference between the path counting and MERW problems more clear, see 3 paragraphs around new formulae (5)-(7) and also the answer to the point (1) of the first referee. We also changed the title of the paper, replacing "maximal entropy random walk" with "path counting" to avoid misunderstanding, and changed terminology in several other places throughout the paper. We also added the reference suggested by the referee.
(2)
Second of all, the maximal entropy random walk on Cayley tree was studied analytically in
Z. Burda and J. Ochab, Phys. Rev. E 85, 021145 (2012), https://arxiv.org/abs/1201.1420
What is interesting there is no sign of the critical value p_cr = p(p-1), instead the solution changes behaviour at p_cr = 2p.
This is a very interesting note and we are extremely grateful to the referee for pointing out this paper to us. We added this paper as reference 13 and included the discussion of the difference between the transition points p_cr^{PC} = p(p-1), and p_cr^{MERW} = 2(p-1) into the Introduction (1st paragraph on page 3). (minor note: k in the notation of Burda and Ochab corresponds to (p-1) in our notation, so their transition happens at 2(p-1), not at 2p).